# The Random Vibrations of the Active Body of the Cultivators



**Petru Cardei** [1]**, Nicolae Constantin** [2]**, Vergil Muraru** [1,*]**, Catalin Persu** [1]**, Raluca Sfiru** [1]**, Nicolae-Valentin Vladut** [1]**, Nicoleta Ungureanu** [2,*]**, Mihai Matache** [1]**, Cornelia Muraru-Ionel** [1]**, Oana-Diana Cristea** [1] **and Evelin-Anda Laza** [1]

[1] National Institute of Research—Development for Machines and Installations Designed for Agriculture and Food Industry—INMA, 013811 Bucharest, Romania; petru_cardei@yahoo.com (P.C.); persu@inma.ro (C.P.); raluca_sfiru@yahoo.com (R.S.); vladut@inma.ro (N.-V.V.); matache@inma.ro (M.M.); cornelia.muraru.ionel@gmail.com (C.M.-I.); diana10.cristea@gmail.com (O.-D.C.); eveline_anda@yahoo.com (E.-A.L.)

[2] Department of Biotechnical Systems, Faculty of Biotechnical Systems Engineering, National University of Science and Technology Politehnica Bucharest, 060042 Bucharest, Romania; nicu.constantin49@yahoo.com

\* Correspondence: virgil.muraru@gmail.com (V.M.); nicoleta.ungureanu@upb.ro (N.U.); Tel.: +40-7-4435-7250 (V.M.)

**Abstract:** The article continues the exposition of the results obtained in researching an agricultural machine for processing soil, designed for research with applications including exploitation. The MCLS (*complex machine for soil tillage*) was designed to research the working processes of the instruments intended for soil processing. The MCLS cultivator is a modulated machine (it can work for three working widths: 1, 2, and 4 m, with tractors of different powers) that is designed to use a wide range of working bodies. The experimental data obtained with the structure with a working width of 1 m and the results of their processing within the framework of the theory of random vibrations are presented in this article. The experimental results are analysed as random vibrations of the supports of the active working bodies. As a result, the main characteristics of random vibrations are exposed: the distribution function, the average value, the autocorrelation, and the frequency spectrum. These general results regarding random vibrations are used for several critical applications in the design, execution, and exploitation of some subassemblies and assemblies of agricultural machines of this type. The main applications include estimating the probability of the occurrence of dangerous load peaks, counting and selecting the load peaks that produce fatigue accumulation in the material of the supports of the working bodies, identifying some design deficiencies or defects in the work regime, and estimating the effects of vibrations on the quality of soil processing. All of the outcomes are composed of applications in MCLS research and exploitation. The applications pursue well-known objectives of modelling the working processes of agricultural machines: safety at work, increasing the quality of work, optimising energy consumption, and increasing productivity, all in a broad context to obtain a compromise situation. The material and the method are based on experimental data acquisition, processing, and interpretation.

**Keywords:** random; vibrations; tillage; tools; complex; cultivator

## 1. Introduction

The phenomena and work processes encountered in agriculture involve, for the most part, important areas of the field of biology (environment, soil, plants, animals, etc.), which is "living". According to all the assessments in the literature, the field of life is a field of unpredictability, uncertainty, randomness, and probability. According to [1], "understanding randomness is essential for modern life, as it underpins decisions under uncertainty".

Random phenomena are treated in physics through theories different from deterministic ones, using the concepts of probability theory and statistics. In [2], it is shown that "in mechanical engineering, random vibration is motion that is non-deterministic, meaning that future behaviour cannot be precisely predicted. The randomness is a characteristic

of the excitation or input, not the mode shapes or natural frequencies. Some common examples include an automobile riding on a rough road, wave height on the water, or the load induced on an airplane wing during flight. Structural response to random vibration is usually treated using statistical or probabilistic approaches. Mathematically, random vibration is characterized as an ergodic and stationary process".

In [3], it is explained that an experiment is random when the answer cannot be precisely and unambiguously predicted, as several answers are possible.

According to [2], the main way of approaching random vibrations is through "a measurement of the acceleration spectral density (ASD), which is the usual way to specify random vibration. The root mean square acceleration (Grms) is the square root of the area under the ASD curve in the frequency domain. The Grms value is typically used to express the overall energy of a particular random vibration event and is a statistical value used in mechanical engineering for structural design and analysis purposes." Additionally, in [2], it is stated that "while the term power spectral density (PSD) is commonly used to specify a random vibration event, ASD is more appropriate when acceleration is being measured and used in structural analysis and testing." Such specifications and assessments can be found in [4–9]. In the case of the study whose results are presented in this article, we worked with sequences of specific deformations measured and recorded, which were then converted into loading forces on the supports of the active working bodies.

As the scientific literature shows, work processes in agriculture have random characteristics [10–20]. Much of the specified literature has agro-economic or bio-economic origins, many of which emphasise management. Uncertainties in agricultural work processes are also caused by management system uncertainties [21]. In [22], it is stated that "problems related to agriculture are, in essence, stochastic because of the uncertain nature of their parameters. Many systems in this sector are affected by yield uncertainty caused by factors such as climatic conditions. Uncertainty and imperfect information involved therein challenge decision-making, as decision-makers are led to make decisions before observing the realization of the random factors. Traditional approaches to dealing with agricultural problems do not integrate the risks and uncertainties involved therein, while it is relevant for efficient managerial decision-making to consider uncertainties and respond to opportunities and threats." The authors [23] show that "the second law of thermodynamics states that entropy or randomness in a given system will increase with time". This is shown in science, "where more and more biological processes have been found to be independent" and "randomness is the fundamental and overarching principle that helps to explain how traits are independently passed from parent to offspring". It is the presence of randomness in all biological systems that this paper aims to highlight.

In the exploitation of agricultural machines, the study of vibrations is carried out more and more experimentally. The results are processed using mathematical statistics in the spirit of the theory of random vibrations [24–30]. The classical theory of vibrations cannot effectively study the complex vibrations of agricultural machines due to objective reasons: the difficulty of estimating model constants, nonlinear behaviours, and excitations, most often of a random nature. Apart from these aspects, the influence of games with a functional character, which is impossible to model in a deterministic way, is manifest. Concrete cases are harvesters [24–26], farm tractors in transport or various agricultural works [25,27–33], as well as cultivator subsoilers [34–36].

In the field of agricultural processes that directly involve the soil, the random character of the process is due for the most part to the random properties of the soil. In agricultural machines intended for soil processing, the soil-working body contact excites the analysed structures. The highly random properties of the soil induce the excitation of the process (especially contact forces) to have a deep random character. In general, the dynamic processes produced by the contact between the soil and the working body of agricultural machines intended for soil processing are well included in the category of random vibrations. Sometimes the machines themselves include vibration sources with adjustable, programmable characteristics to achieve a certain type of soil processing. In

the latter case, in the experimentally recorded signals, it is natural for the deterministic vibration elements to appear in the spectrum, even if they have been altered to some extent. The random vibrations of agricultural machines intended for soil processing are another example of random vibrations encountered in the technique, apart from those given in [2]. We recall the definition of the agricultural cultivator as given in [37], that is, an agricultural machine that serves to shred and loosen the soil, to destroy weeds from crops of creeping plants, etc. This definition was added because the study material presented in this article has as a main component a working variant of a complex cultivator with variable working width. According to [38], the cultivator can be understood as an agricultural machine used for surface soil work to loosen and destroy weeds without overturning the furrow.

The original aspect of this study is the presentation of a research method based on the theory of random vibrations using the capabilities of a complex modulated cultivator with a variable working width, MCLS. MCLS performance testing is presented simultaneously as a research method for other similar machines or for machines obtained from MCLS by modulating or fitting different active bodies that the structure can use.

Also, the paper tries to show the researchers the benefits produced by the study of vibrations in an experimental framework with the help of the theory of random functions. Most researchers are trained in the scientific spirit of the deterministic theory of vibrations, which is why their first tendency is to look for answers in terms of the deterministic theory. From there, it comes to mathematical modelling and simulation, which are easy to develop on the computer. After the models are run, however, it is found that it is difficult to give answers to the real problems, especially to motivate and validate the hypotheses and mathematical models. Advanced designers in the field of such phenomena have been using the path of experimentation and the theory of random functions for decades. The deterministic approach tries to force physical reality into the model's pattern, while modelling in the experimental framework seeks to take as much as possible from the real conditions and extract a minimum of useful conditions in conception, design, execution, and exploitation. The most difficult problems with this last option are the generalisation (which is low for the time being) and the generalisation costs (which are very high).

## 2. Material and Methods

Research on the MCLS soil tillage machine (a variant with a working width of 1 m) is the subject of this article. The presented results describe the experiments and the statistical processing of the experimental data. The MCLS complex cultivator was initially conceived, designed, and realised to facilitate research in the field of soil processing [39,40]. The machine was designed for three working widths: 4 m and 2 m (with an 80 HP tractor) and 1 m (for a 45 HP tractor). Additionally, the supporting structure can support the installation of 5–10 types of working bodies to make comparative estimates as complete as possible. After conception, design, and execution, the idea of using the machine in farm operations also appeared. Thus, the machine can be used simultaneously for farm applications and research. Images from the experimental activity with the MCLS machine in all working variants are given in Figures 1–4.

The experiments were carried out in Romania, in the Bucharest-Ilfov area, in an experimental plot owned by the INMA Bucharest Institute, which has a transitional temperate continental climate. The morphological characteristics of the soil were: red-yellow clay loam, in a dry state; medium and small angular agglomerate structures; slightly loosened. The soil moisture at the measurement time was between 20 and 35 percent (Delta-T HH2 digital moisture meter), while the average soil compaction level was 45 KPa.

Traction is provided by agricultural tractors New Holland 80 TD and New Holland 50 TD. The working speeds were between 0.78 and 2.15 m/s. The results presented in this article refer to the experiment carried out at a constant speed of 0.789 m/s.

The experiments were carried out on a plot of 500 square metres, marked with benchmarks. For each test, 10 m were set for the acceleration and 10 m for the deceleration of the tractor, and 30 m were set for the evaluation of the cultivator (see Figure 4).

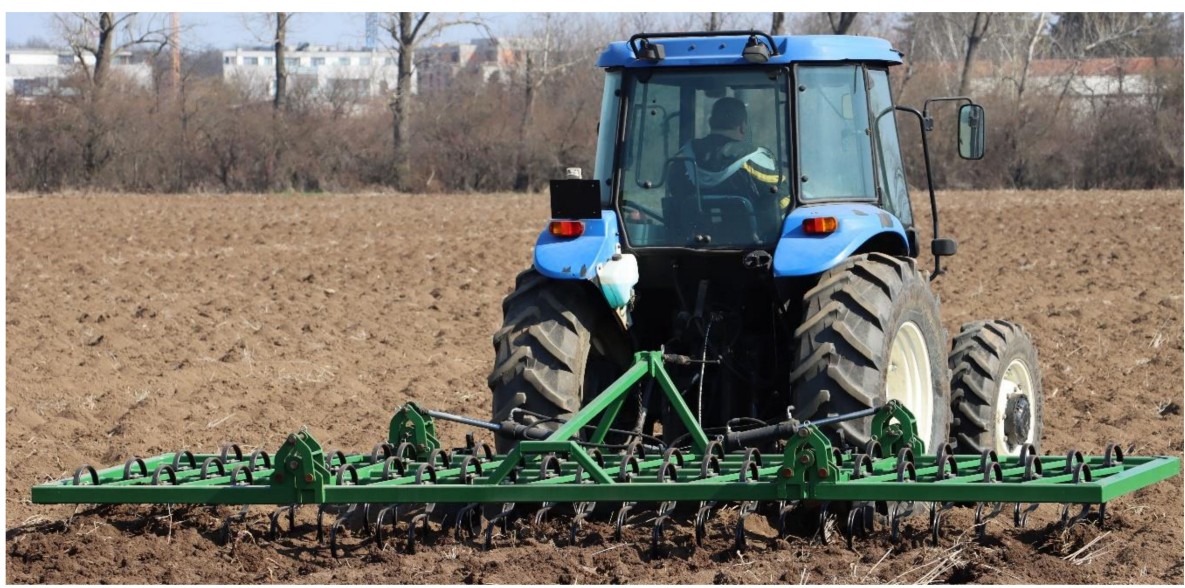

**Figure 1.** The MCLS complex cultivator operates in the version with a working width of 4 m.

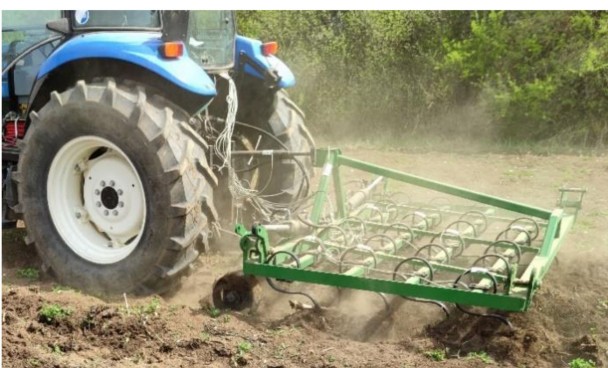

**Figure 2.** MCLS in operation in the 2 m working width variant.

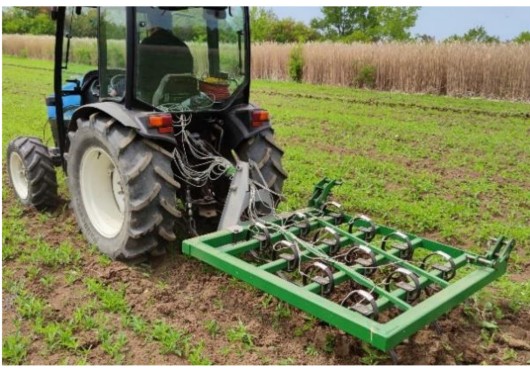

**Figure 3.** MCLS in operation in the 1 m working width variant.

The QuantumX MX1615B data acquisition system was used for the strain gauge measurements, with 32 measurement channels and KFG-6-120-C1-11 N15C2 strain gauge sensors (120 $\Omega$). The strain gauge amplifiers, QuantumX MX1615B, are suitable for precise data acquisition in full-bridge, half-bridge, and quarter-bridge configurations, as well as for strain gauge-based transducers, potentiometers, resistance thermometers (PT100), or normalised voltage ($+/-10$ V).

The measurement of the forces on each support of the working body was performed by the strain gauge methodology, with the procedure including a calibration stage of

the indications of the deformation sensors. Although it is not the most precise measurement method, it is convenient and accessible, especially considering that only values measured by the same procedure are compared and are, therefore, hypothetically, affected by comparable errors.

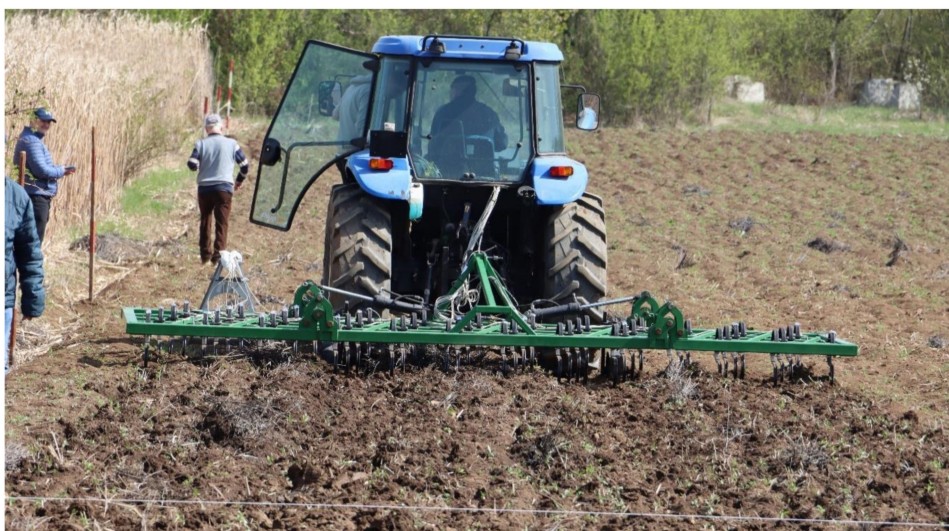

**Figure 4.** The location and parcelling of the land on which the experiments were carried out with the MCLS machine.

The experiments took place in the testing ground of INMA Bucharest, where the soil was classified as reddish-brown from the forest and pre-processed with autumn ploughing (coarse). Figure 5 depicts the location of the experimental polygon in detail with reference to the Bucharest-Baneasa airport runway.

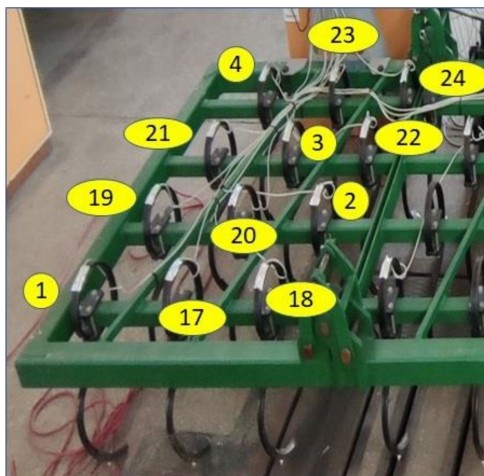

**Figure 5.** Indexing of measurement locations on the structure.

### 2.1. The Subject of Research

Approaching the soil processing phenomenon using an MCLS-type structure involves the statistical processing of experimental results (descriptive and inferential statistics, random process theory). Descriptive and partial inferential statistics were discussed in a previous article [41]. In this article, the focus will be on approaching the problem within the theory of random processes. In another article [41], all the data strings that are generated by the contact between the ground and the working body have a random character and are comparable to the pseudorandom strings generated by special programmes or to the strings of prime numbers of the same length.

In order to approach it within the theory of random functions, the phenomenon of ground processing by each organ was considered a phenomenon of random vibrations. The main random component is the excitement of the process. Excitation is generated by the interaction of the working body with the ground. In the range of demands considered, the vibrations can be considered linear-elastic.

In this article, the analysis will be made only for the MCLS variant with a working width of 1 m because, in the case of this variant, signals were collected from all the supports of the working bodies throughout the experiment. The analysis carried out on the simple version with a width of 1 m can later be extended to the versions with higher working widths, for which the load on each support cannot yet be measured and other random parameters also appear (for example, the clearances in the truss with a working width of 4 m).

The indexing of the measurement locations and the signal transmission channels from the deformation sensors to the acquisition board are shown in Figure 5. An image of the MCLS variant in experimental work, on the parcelled track of the polygon, with the complete data acquisition equipment, is shown in Figure 6.

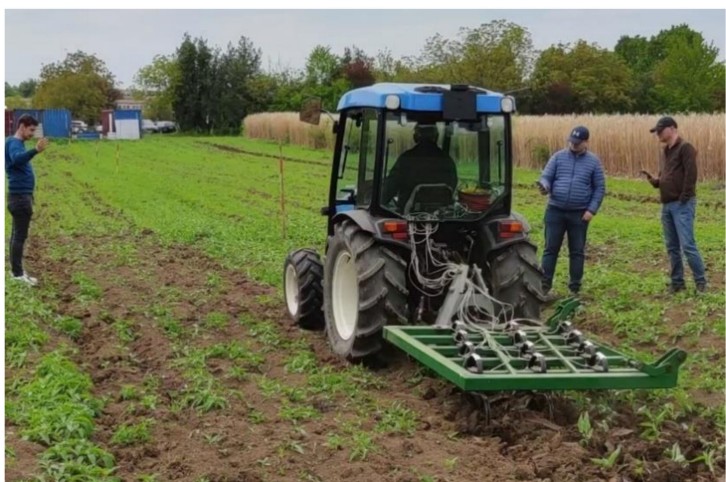

**Figure 6.** MCLS variant with a working width of 1 m in operation with a 45 HP tractor.

2.1.1. The Formulation of the Problem in Terms of the Theory of Random Vibrations

According to [42], physical processes that cannot be characterised by deterministic functions, i.e., vibrations whose instantaneous values cannot be predicted as functions of time, are called random. The random nature of the vibrations of the investigated structure was demonstrated in [41], where even the degree of randomness was quantified. A relatively simple criterion for experimental recognition of the randomness of a phenomenon that is experimental is formulated, for example, in [42]: if, in several experiments organised under identical controllable conditions, the measured quantities differ only by quantities of the order of errors of measurement, the phenomenon can be considered deterministic; it is reproducible, so its development after the moment of measurement is predictable. Otherwise, the phenomenon is random. Apparently, the criterion seems simple, but in the case of soil processing and, in addition, using non-conventional and insufficiently verified structure load measurement systems (strain gauge methodology), this criterion is difficult to use. Given the findings from [41], we opted for the random vibration variant to describe the working process of the 1 m working width version of the MCLS complex cultivator.

The main characteristic of a random variable is the distribution function [43,44]. This is defined according to Formula (1).

$$F_X(x) = P(X < x), \ x \in \mathbb{R} \tag{1}$$

where $F_X$ is *the distribution function of the random variable x, x is a real number, and* $\mathbb{R}$ *is the* set of real numbers. In Formula (1), some authors use non-strict inequality.

### 2.1.2. Distribution Functions

In Figure 7 in the column of graphs on the left, the temporal variations of the forces that load the supports of the twelve working bodies of the version with a working width of 1 m of the MCLS are shown. Additionally, in Figure 7, the histogram of the frequencies of the force values is represented in the right column. The polygon of frequencies is built using the histograms. When the number of classes in the histograms becomes very large (theoretically to infinity), the frequency polygon tends to reflect the real distribution density of each force. The frequency histograms are constructed with 64 classes using the formula of Mosteller and Tukey [45]. We took into account the length of the data strings obtained experimentally in calculating the number of classes in the histograms.

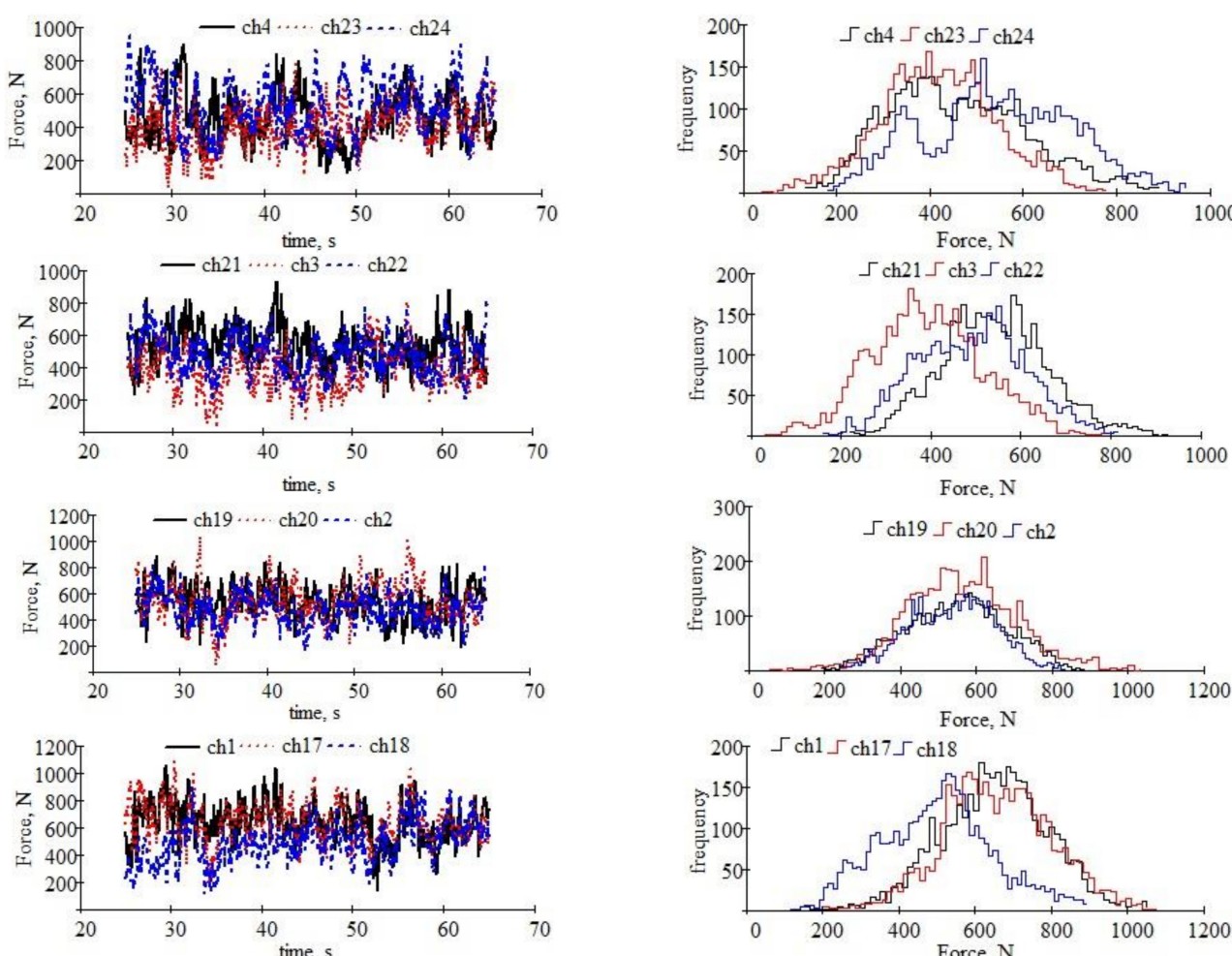

**Figure 7.** The time dependence of the load in the supports of the active bodies (**left column**) and the frequency distribution of the load values (**right column**). The four rows of figures correspond to the four lines of active organs, starting from the top with the ones closest to the tractor.

The elementary Formula (2) was used to calculate frequency histograms.

$$f_i = \sum_{j=1}^{N} H_i(x_j), \ i = 1, \ldots, n_h, \ j = 1, \ldots, N \tag{2}$$

where:

$$H_i(x_j) = \begin{cases} 1, h_i \leq x_j < h_{i+1} \\ 0, x_j < h_i \ or \ x_j \geq h_{i+1} \end{cases}, \quad i = 1, \ldots, n_h, \quad j = 1, \ldots, N, \tag{3}$$

$$H_{n_h}(x_j) = \begin{cases} 1, h_{n_h-1} \leq x_j \leq h_{n_h} \\ 0, x_j \langle h_{n_h-1} \ or \ x_j \rangle h_{n_h} \end{cases}, \quad j = 1, \ldots, N \tag{4}$$

$N$ is the number of observations from the examined random sequence, and $n_h$ is the number of classes in the frequency histogram. Then the probability density histogram was calculated according to Formula (5):

$$\rho_i = \frac{f_i}{N}, \ i = 1, \ldots, \ n_h \tag{5}$$

where $\{\rho_i\}_{i=1,\ldots,n_h}$ is the series of probability densities. The series of probabilities (or cumulative probabilities) is calculated using Formula (6).

$$p_i = \sum_{k=1}^{i} \rho_k, \quad i = 1, \ldots, n_h \tag{6}$$

In Figure 8, the probability density and the probability that the random variable (the series of numerical values over time recorded at each of the twelve measurement locations) will take different values in the experimental range are shown.

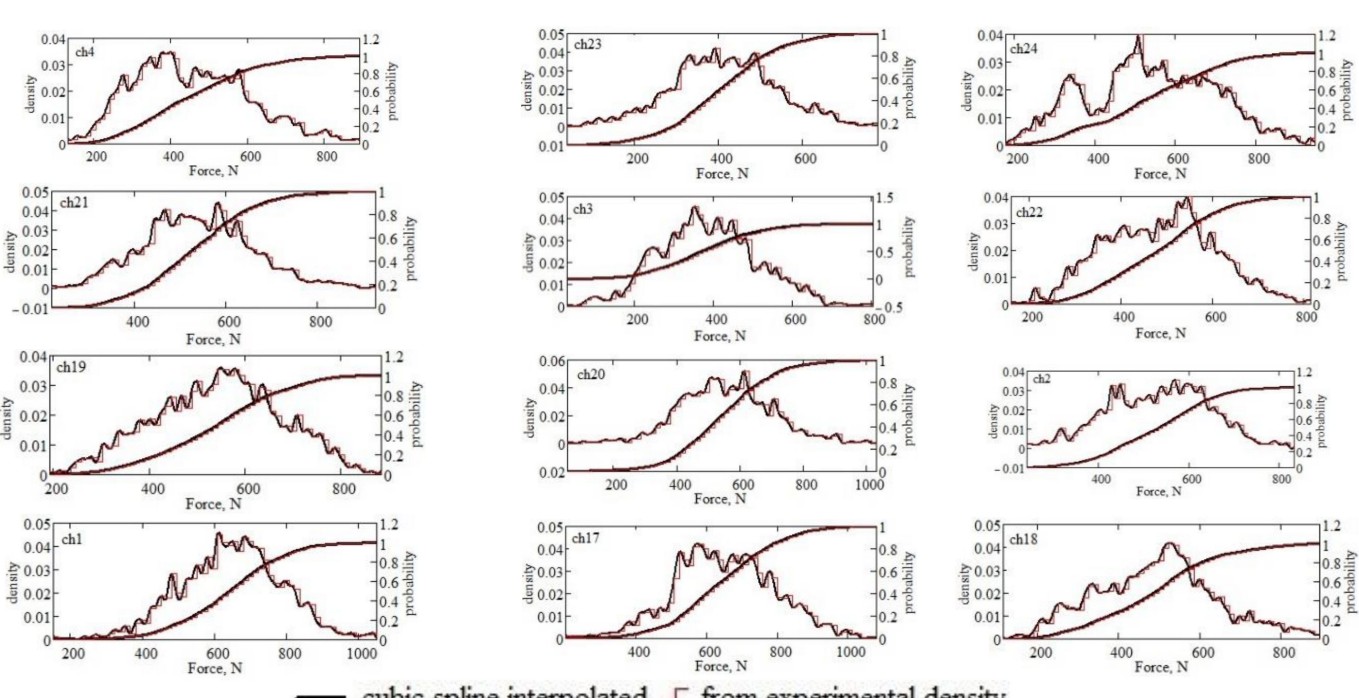

**Figure 8.** The probability density and the probability of taking certain values from the experimental range for the forces recorded in the twelve measurement locations.

The probability densities calculated as shown above do not resemble those found in statistics books or in articles in which the experimental results are approximated, by hypothesis, with one of the classical statistical distributions (normal, exponential, Student, Fischer, etc.). Probability distributions differ from the statistical densities with which experimental curves are usually approximated. Precision is lost by approximating with the classical statistical distributions of the probability densities, and we cannot precisely estimate the local error in relation to the experimental data. For this reason, we preferred, considering our goals, to interpolate the empirical probability densities through cubic spline

functions since we wanted maximum precision and were not interested in using the results in immediate theoretic models. Regarding the probability curves, they have shapes very similar to the classical ones, but for the same reasons, we proceeded with the probabilities by interpolating with spline functions. In Figure 8, the experimental curves interpolated by spline functions are given in each graph for the twelve series of 4000 numerical data points.

The probabilities interpolated with spline functions are used to calculate the probability of the occurrence of a force greater than a fixed force (related to the limit resistance characteristics of the material of the supports of the working body).

The distribution functions of the twelve random variables described by the numerical sequences in Figure 7 are calculated in the same way as the probability densities and probabilities whose graphic representations are given in Figure 8. Therefore, the distribution functions are expressed as interpolations by cubic spline functions, for which the graphic representations are given in Figure 9. The graphic representations of the twelve distribution functions of the sequences of numerical values coming from the measurement points located on the supports of the twelve working bodies are grouped three by one in the graph, corresponding to the three working bodies in each line, with the counting starting from the back of the tractor.

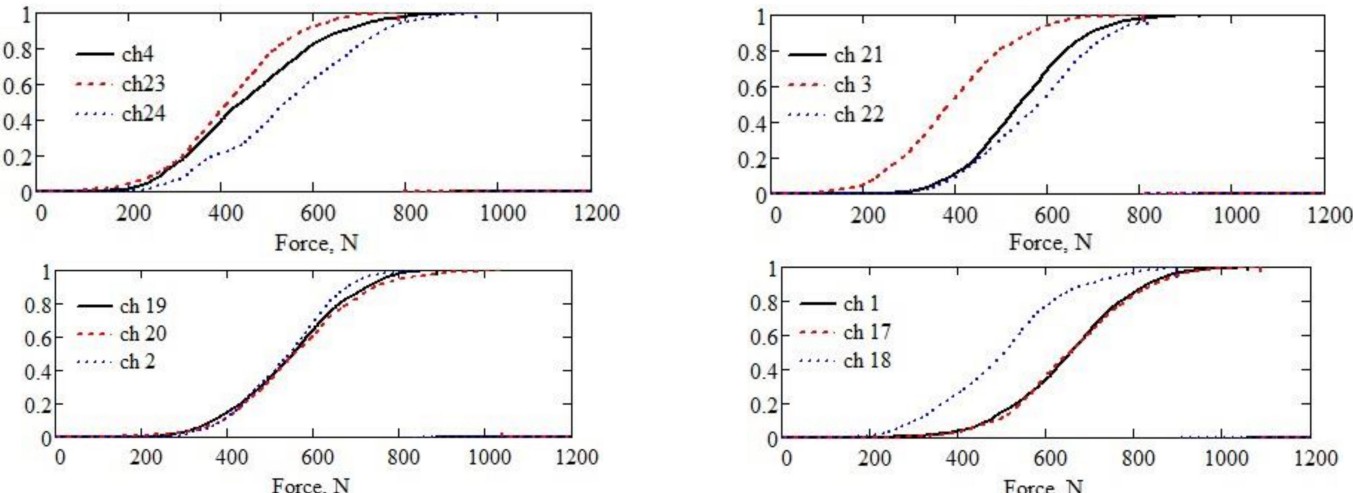

**Figure 9.** The graphic representation of the distribution functions of the twelve numerical data sequences was obtained by measuring at the locations indicated in Figure 5.

For complete agreement with the theoretical definition, the interpolated distribution functions are extended with the value zero to the left of the minimum value and with the value one to the right of the maximum value of the corresponding random variable.

### 2.1.3. Characteristic Functions of Random Vibrations

From a theoretical point of view, the first tested properties of random vibrations are *stationarity* and *ergodicity*. The definitions of these notions can be found in all the literature dedicated to random vibrations, for example, in [2,4–9,42,43,46]. In general, the response of structures to random vibrations is studied through experiments that have strictly elaborated procedural standards [47].

A synthetic characterisation of random signals is given in [3]. The non-deterministic signal has specific characteristics: mean, dispersion, global average, global dispersion, histogram, power spectral density, etc. The signal can have a certain degree of predictability in its evolution over time. Depending on specific characteristics, the non-deterministic signal can be:

*Stationary*, if the mean and dispersion do not depend on time but are constant;

*Ergodic*, if the mean per portion does not differ from the global mean;

*White noise*, with a constant power spectral density throughout the frequency band.

Testing the stationarity and ergodicity of the experimental signals (sequences of finite length) cannot be conducted using the definitions of the mean value and the autocorrelation function of the random variables, for example, in [42]. The definitions in [42] assume processes for crossing the limit after the number of samples. We do not have an infinite number of samples, and if we did, we would be constrained by costs to limit the duration of the experiences as much as possible. For this reason, the calculations are made for a finite number of samples, limited by the number of samples in the entire sequence.

In Figure 10, we see the behaviour of the average value of the random variable given by the registration with the ch4 code (coming from the working body in the first line after the tractor, on the extreme left). The tendency to decrease the amplitude is observed with an increase in the number of samples in the sequence considered for calculating the average value. Therefore, one can only suspect asymptotic behaviour, but such behaviour cannot be stated strictly theoretically.

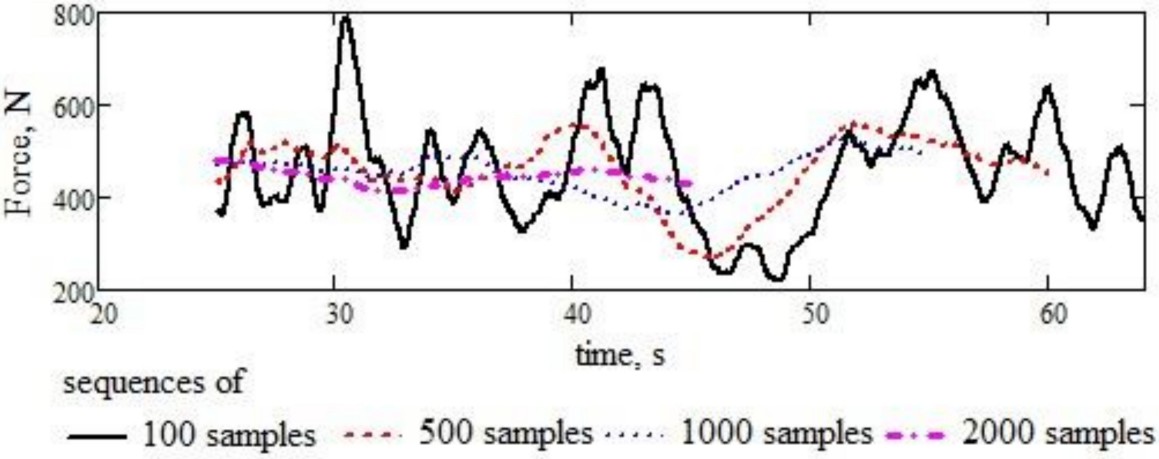

**Figure 10.** The behaviour of average values increases as the number of samples used in their calculation increases.

In Figure 11, the terms below the limit of the definition of the autocorrelation function are calculated for the subsequence of the ch4 signal, with the graphic representation limited to only five values of the gap. There is a tendency to decrease the amplitude of the autocorrelation function with the gap value and a weak asymptotic tendency, which does not allow us to make statements about the stationarity and/or ergodicity of the signal.

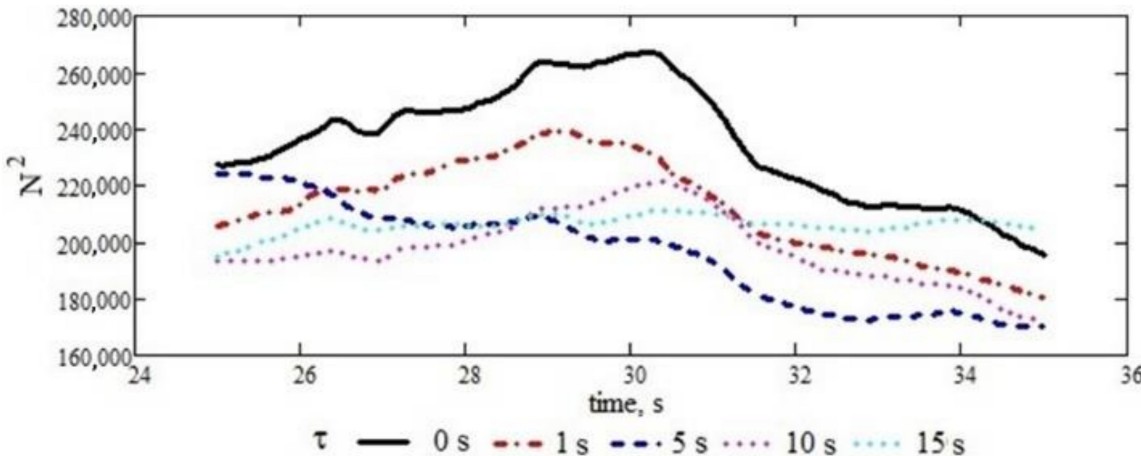

**Figure 11.** Autocorrelations of the ch4 signal for five offset ($\tau$) values.

In Figure 12, the autocorrelation curves are given for each of the twelve signals collected from the supports of the working bodies of the 1 m working width version of the MCLS. The curves are calculated using the *lcorr* function of the programme [48]. The calculation can also be performed directly by programming the three simple formulas given in [49]. These curves can be considered substitutes in the field of real signals of finite length for the assessment of the stationarity or ergodicity of signals. In Figure 12, it is first observed that all twelve signals tend to asymptote in time towards the value 0. First, this means that the signals are weakly autocorrelated (another argument for their randomness). The fact that they are not constant in time shows that the signals are neither stationary nor ergodic.

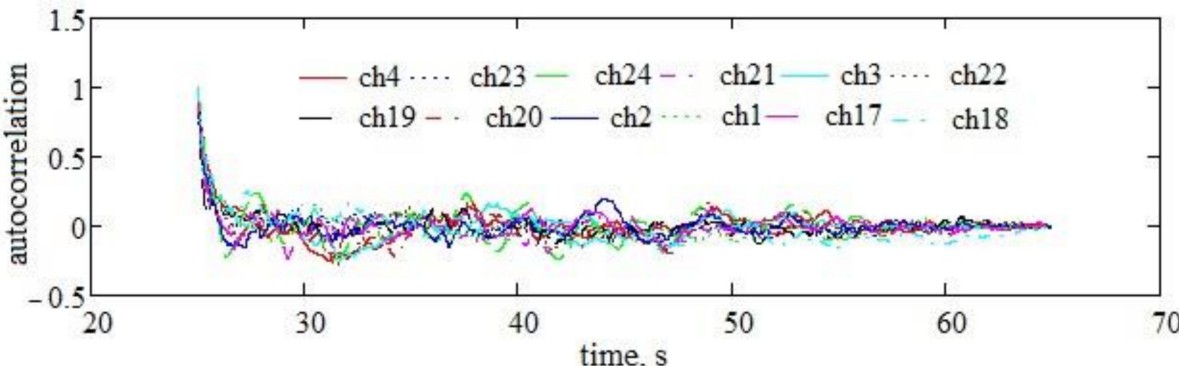

**Figure 12.** The autocorrelations of the twelve signals recorded in the experiment were calculated using [50].

### 2.1.4. Spectral Analysis

The frequency spectrum of the signals collected from the twelve working bodies provides interesting information, especially from identifying possible deterministic components that appear in all the examined sequences, which indicate possible deficiencies of the working regime, abnormal clearance, or defects. Deterministic components can also appear in agricultural soil processing machines that produce programmable vibrations, which is not the case for the structure examined in this paper.

Another application of the frequency spectrum of the random sequence is the estimation of the risk of resonant vibrations produced by random components on the elements of the cultivator structure. Components of the excitation that produce resonances are unlikely because the coincidence of a constant excitation with one of the first natural frequencies of the structure (here we refer only to the support of the working organ; in the general case, the calculation is very voluminous) is difficult to manifest. Even if it happens, the damping from the metal material and the ground on which it moves severely diminishes the resonant effect. Therefore, resonance effects are probably due to the improper functioning of some internal organs or a work speed that, combined with the soil profile, can produce them. However, a comparison between the spectrum of the supports of the working body and the spectrum of the excitations can be made to investigate the resonance problem at the supports of the working body or some more complex sub-assemblies. Figure 13 shows an example of a study of the distribution of the natural frequencies of the studied structure (here, the supports of the working body) in relation to the excitation spectrum (signals from the supports), obtained using the Fourier transform. In Figure 13, with solid grey bars, the spectra of the twelve supports of the working bodies are drawn in the 14–16 Hz range, in which the fundamental frequency of the support falls. The red bar with a circle at the upper end marks the position of the fundamental natural frequency of the supports in the chosen frequency range. The fundamental Eigen frequency and the other Eigen frequencies are calculated and shown in Figure 14 (the first five natural frequencies), but they were not experimentally determined as they should have been when calibrating the structure.

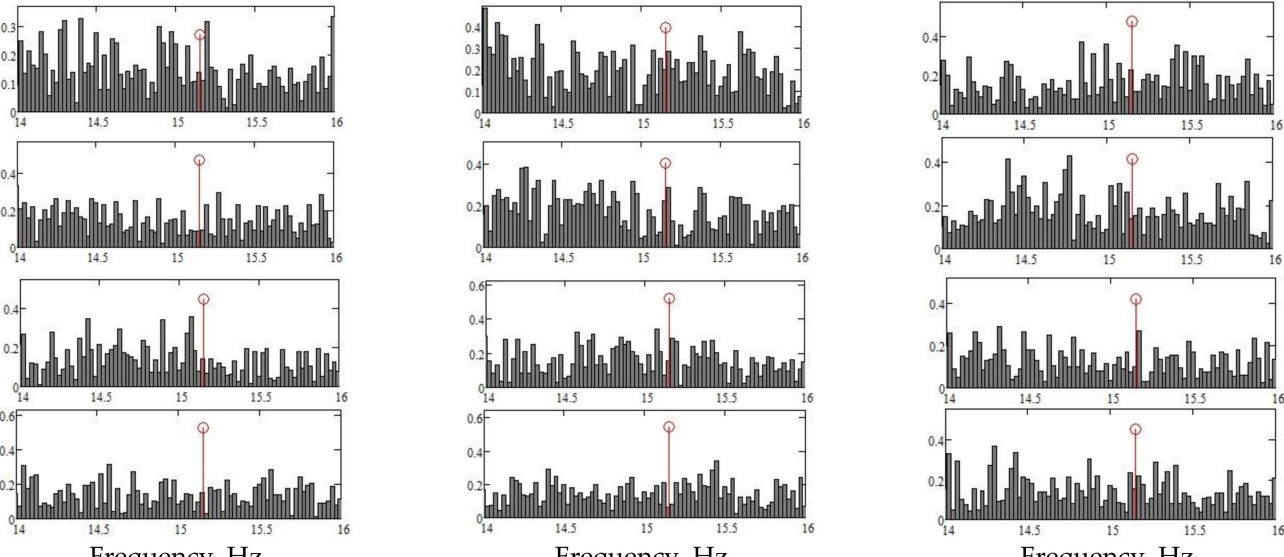

Frequency, Hz  Frequency, Hz  Frequency, Hz

**Figure 13.** Representation of the frequency spectrum of the twelve signals from the supports of the working bodies, compared to the fundamental frequency of the support, calculated by the finite element method (represented by the thin red bar with a circle at the upper end).

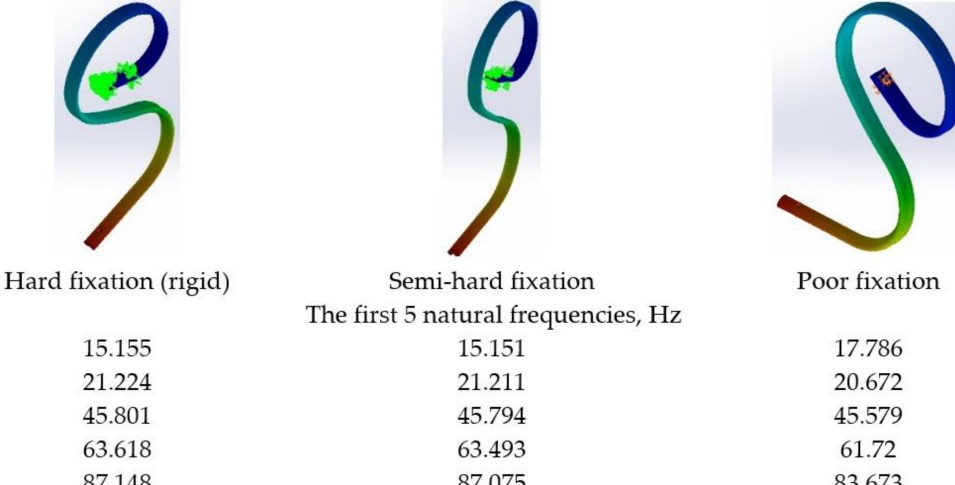

| Hard fixation (rigid) | Semi-hard fixation | Poor fixation |
|---|---|---|
| | The first 5 natural frequencies, Hz | |
| 15.155 | 15.151 | 17.786 |
| 21.224 | 21.211 | 20.672 |
| 45.801 | 45.794 | 45.579 |
| 63.618 | 63.493 | 61.72 |
| 87.148 | 87.075 | 83.673 |

**Figure 14.** The first five natural frequencies of the supports of the working bodies of the MCLS in three variants of fixation on the load-bearing structure and the deformed forms of the structure in the vibration mode correspond to the fundamental frequency.

Since we had no data about the operation in the non-linear elastic or plastic domains or about the damping capacity of the steel from which the supports are built, we worked in the linear elastic domain, which makes the displacement values exaggerated when calculating the natural frequencies. For this reason, we have not given numerical data relative to the deformed shapes corresponding to the five fundamental natural frequencies.

The value of the fundamental natural frequency represented by the red bar with a circle at the upper end in each of the twelve graphs in Figure 13 corresponds to the support or semi-hard fixation of the support (15.151 Hz). Overlaps of the fundamental frequency over local maxima of the excitation spectrum occur for channels ch4, ch23, ch24, ch3, ch20, ch2, and ch1. However, there are no reasons for concern because the peaks in the spectra of the supports have small values and the damping capacity of the material of the supports, and especially of the soil in which the working body moves, is high.

In addition, when calculating the spectra of all the twelve signals coming from the twelve measurement locations, it is found that they are all highly correlated, with the

minimum value of the correlation of two such numerical sequences being 0.979. The most intensely correlated signals correspond to channel pairs: ch2 and ch17, ch2 and ch1, and ch1 and ch21. At the lowest intensities, we find correlations between channels ch4 and ch1, and between ch4 and ch17.

A better understanding of the distribution of the frequency spectrum of all twelve signals coming from the measurement location of the supports of the working body of the cultivator in relation to the natural (calculated) frequencies of the supports is possible using the graphic from Figures 15 and 16.

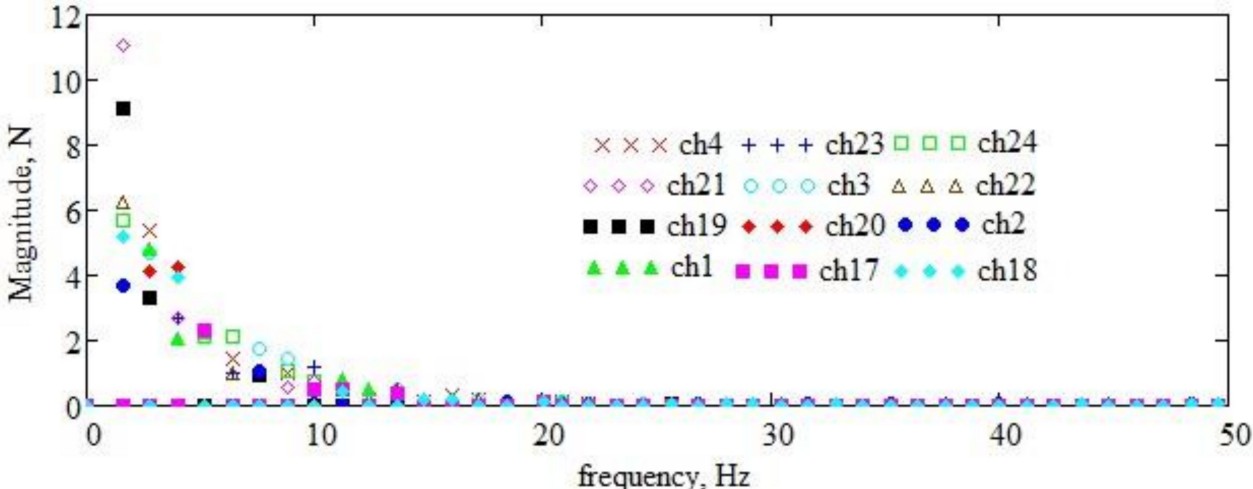

**Figure 15.** The frequency spectrum for the twelve signals (in the 0–50 Hz window).

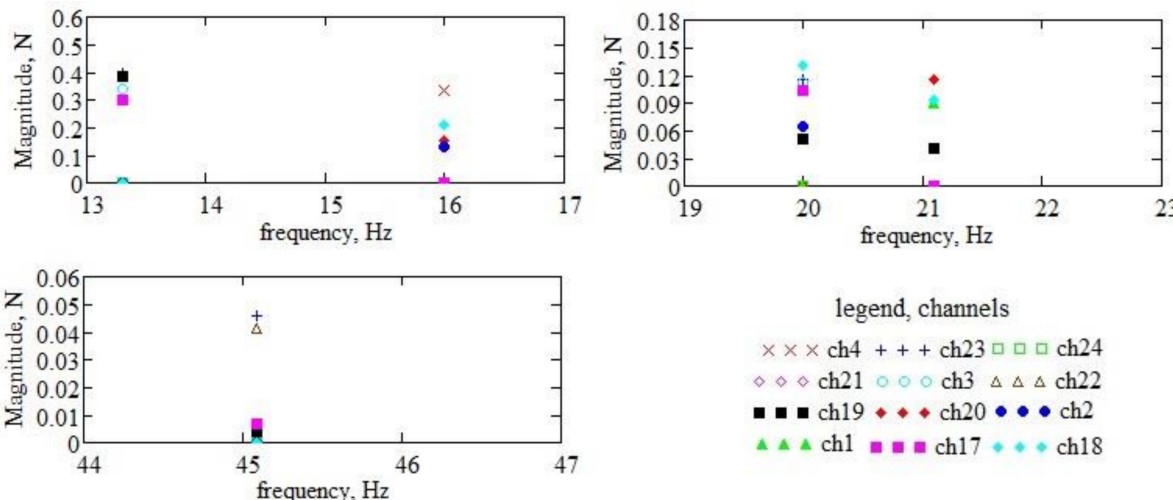

**Figure 16.** The frequency spectra of all twelve signals in narrow windows around the first three natural frequencies of the support of the working organ (see Figure 14).

In Figure 15, the spectral distribution is plotted for all signals in the range (window) 0–50 Hz, which includes the first three fundamental frequencies. Details in narrow intervals around the first three fundamental frequencies (see Figure 15) are plotted in Figure 16. It is found that, around the first three natural frequencies of the supports of the active organs, there are few frequencies from the signal spectrum, and, in addition, their magnitude is very small (when comparing the magnitudes of the frequencies in Figure 17 with those of the spectrum components in Figure 15).

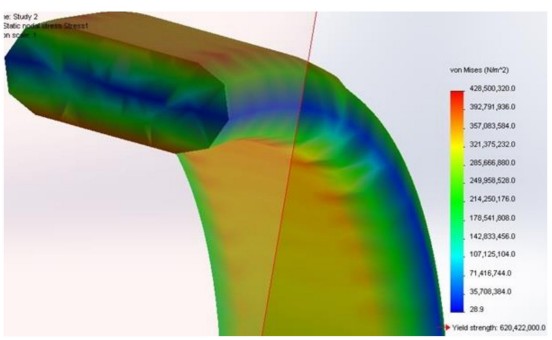 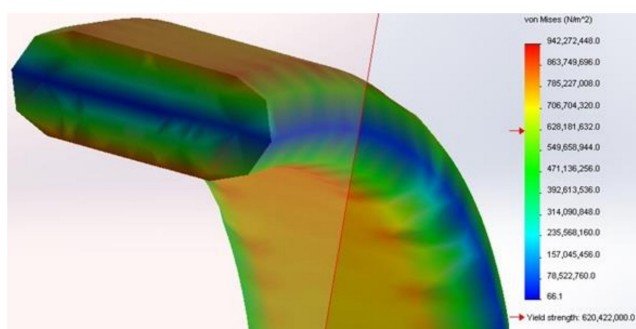

**Figure 17.** The von Mises stress map on the section with the maximum load on the support of a working body of the MCLS. The load values were 518 N (**left**) and 1086 N (**right**).

### 2.1.5. RMS and the Average Value

In many studies of random vibrations, RMS (root mean square) is calculated as a measure of amplitude extremes. The frequent use of RMS in the study of vibrations is also due to its direct connection with the energetic content of vibrations and implicitly with their destructive capacity [51]. RMS is related to the vibration amplitude and its average value.

Table 1 gives the RMS values and the average value for each sequence recorded in the examined experiment. It can be seen that the two measures of the force amplitude are close. Moreover, the correlation between the two matrices (that of the RMS and the average values) is very high: 0.999.

**Table 1.** RMS and average values for the numerical sequences corresponding to the twelve measurement points (see Figure 6), organised in the table according to the order of the physical structure.

| RMS, N | | | Average Values, N | | |
|---|---|---|---|---|---|
| **477.145** | **429.029** | **560.354** | 453.124 | 410.584 | 537.523 |
| 549.272 | 405.641 | 498.193 | 536.507 | 385.602 | 483.493 |
| 557.469 | 571.771 | 543.765 | 541.699 | 552.885 | 532.069 |
| 659.646 | 665.703 | 514.895 | 644.269 | 650.431 | 493.188 |

## 3. Results

The main results that will be presented are applications for the design, execution, and operation of agricultural machines such as the MCLS complex cultivator, which is presented in Section 2. The results refer to the calculation of the probability of the occurrence of dangerous peaks, the selection and counting of the peaks of force that produce fatigue accumulation (in the supports of the active organs), the identification of design defects or deficiencies or of the work regime, and effects on the quality of the work.

### 3.1. The Probability of the Occurrence of Dangerous Loads

To identify and count the dangerous peaks of force in support of the working organ, the characteristic resistance limits of the material from which it is constructed will be used: the bending fatigue limit stress, $\sigma_{-1}$; the yield stress (plasticisation), $\sigma_Y$; and the breaking or yielding limit stress, $\sigma_r$. To be able to solve the proposed problem, the finite element method was used to determine the state of equivalent tension in support of a working body as a result of the application of forces with experimentally determined values. The material used has the following limit characteristics: $\sigma_Y$ = 620.422 MPa, $\sigma_r$ = 723.826 MPa, modulus of elasticity $E$ = 210,000 MPa, Poisson's ratio $\nu$ = 0.28, mass density $\rho$ = 7700 kg/m$^3$, and the fatigue resistance limit by bending, $\sigma_{-1}$ = 380 MPa. The structural model used is described in [16,17]. The occurrence of damage phenomena (irreversible deformations, cracks, or breaks in the material) is determined with the help of the structural model by comparing the values of the equivalent stress (von Mises) in the structure with the critical limit stresses

of the material. In Figure 18, the colour maps of the state of equivalent tension resulting from the application of the average (518 N) and maximum (1086 N) forces to the support of the working body are shown.

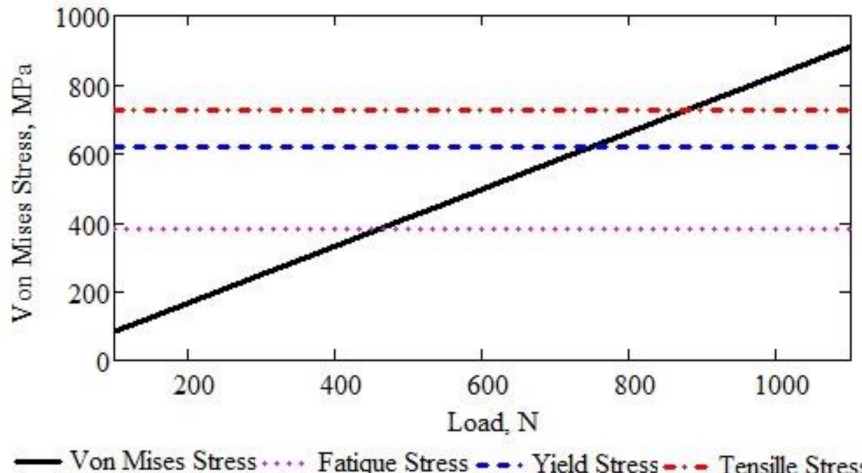

**Figure 18.** The variation in the maximum von Mises stress in the material supporting the active organs depends on the loading of the organ compared to the critical limits of the material.

It can be observed that the maximum equivalent stress for the load of 518 N has a value of 428.5 MPa (with a maximum resultant relative displacement of 77 mm), and for the load of 1086 N, it has a value of 942.3 MPa. Therefore, the average load of 518 N does not pose damage problems, and the support works in the field of linear elasticity with a safety coefficient of approximately 1.45. In the case of the maximum stress (1086 N), the equivalent stress in the structure reaches maximum values higher than the breaking limit stress of the material. However, the organ will not yield yet because the core of the bar from which the support is built presents an appreciable area that works in the linear elastic domain. For the maximum value of 1086 N, the yield stress is exceeded in an appreciable area (Figure 17), and even the breaking stress is exceeded in narrow areas in strips near the longer sides of the cross-section. However, the section works in linear elastic mode in the central area, so a failure is not directly recorded. However, in this working regime, high peaks lead to the accumulation of fatigue and possibly premature failure.

Using the results provided by the structural analysis of the support of the working body and the probabilities interpolated with cubic spline functions, it is possible to calculate the probability of exceeding the limit forces that cause the critical limits of the material to be exceeded: fatigue, plasticisation, and breaking. First, based on the hypothesis of the linear elastic behaviour of the support material of the working body, the graphic representation is shown in Figure 18.

Note: To calculate the equation of the oblique line in Figure 18, we accepted the hypothesis of linear elastic behaviour of the support material upon bending up to the plasticisation limit. Thus, the fatigue limit is included in the elastic bearing. The equation of the oblique line in Figure 18 is:

$$\sigma = Kx \tag{7}$$

where the elastic constant $K$ is calculated using the result of the structural analysis for the average value (518 N) of the force applied on the twelve sequences of analysed force values: $K = 0.827$ N/Mpa.

The probability that the loading force corresponding to each of the twelve supports of the active organs will exceed the critical values of flow, rupture, and fatigue (characteristics of the material of the supports) is shown in Figure 19.

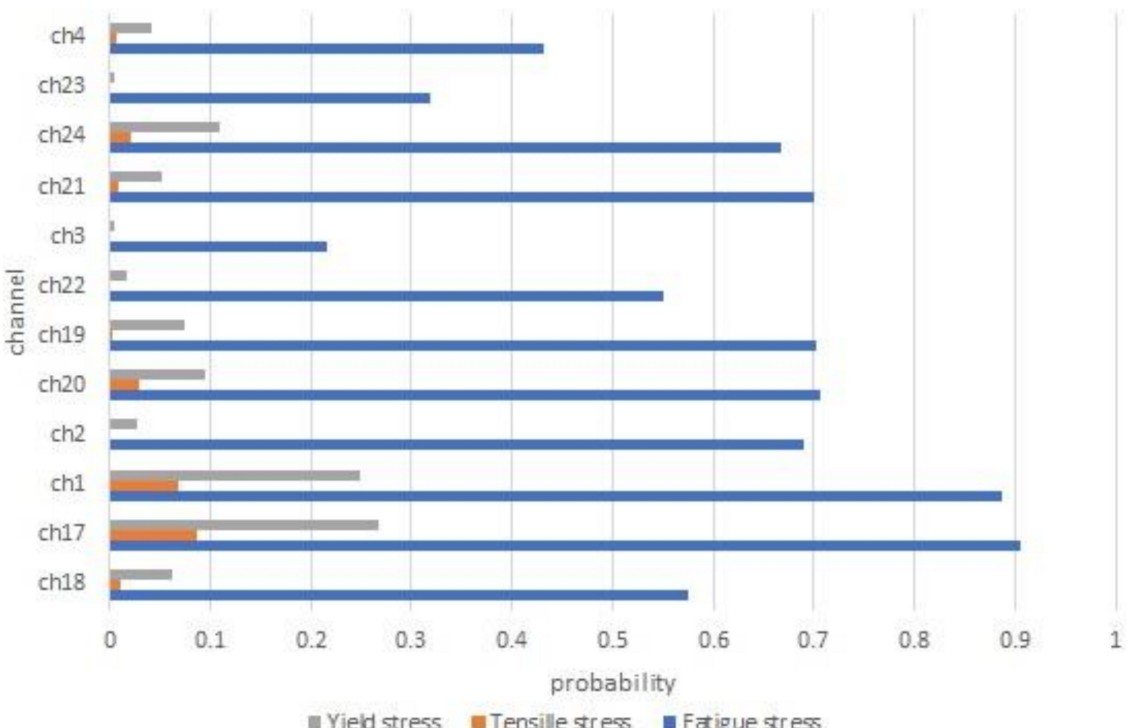

**Figure 19.** The probability that the loading force will exceed the limits of flow, rupture, and fatigue of the material of the support of the working body.

*3.2. Counting the Peaks That Produce Fatigue*

One of the immediate applications of this research is the possibility of counting the "cycles" that produce fatigue in the support material of the working body. Using the data and the graph in Figure 18, one can find that the "part" of the random sequence of force values that requires the support of the working body can produce the accumulation of fatigue, cracks, and breaks in the material.

A stress sequence filter is the simplest selection method for the part of the stress that produces fatigue accumulation, plastic yielding, or breaks. The expression of such a filter is given by Formula (8):

$$\varsigma_i = \begin{cases} s_i, & s_i > \sigma_{-1} \\ \sigma_{-1}, & s_i \leq \sigma_{-1} \end{cases}, \quad i = 1, \dots, N \tag{8}$$

where $s$ is the sequence of experimental data in force values converted into values of the equivalent stress (von Mises) in the material, according to Formula (7).

In Figure 20, the load peaks of the support of the active body from the extreme left of the first line of working bodies (ch4) after the back of the tractor are highlighted. Prominent spikes produce fatigue buildup in the support material.

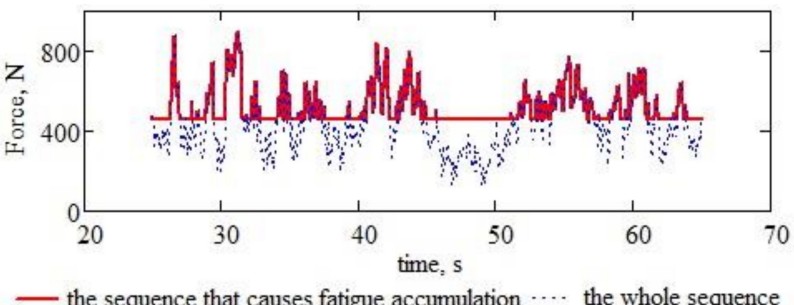

**Figure 20.** Filtering the sequence of forces that produce the accumulation of fatigue in the signal corresponding to channel ch4.

The wear of the support in 40 s will be calculated by structural engineers using the above data. During this time, the vehicle covers approximately 30 square metres of land. The result allows the estimation of the life span in time (neglecting the wear caused by the action of water or other environmental factors) in hectares of ploughed land or seasons or years of exploitation. In order to determine the lifetime of the supports of the working bodies, any structural analysis programme that has this facility can be used (for example, [52,53]).

### 3.3. Identification of Defects, Deficiencies, or Operating Errors

Most often, the study of random vibrations is dedicated to or requires the inclusion of the identification of sources of errors, defects, deficiencies, premature wear, etc. in the operating regime. Measuring the random vibrations of structures and improving or even optimising them based on the measurements has become one of the most commonly used methods in the study of phenomena affected by vibrations [54–58].

The research carried out in the MCLS complex cultivator case highlighted work deficiencies, especially the problem of low-quality control of the working depth. It was shown in [41] that using only two working depth control wheels (located in the front of the MCLS) is insufficient. A pair of wheels would also be necessary at the rear. In addition, the suspension-traction system (connection to the tractor) should also be checked to not raise the structure at the front during work. The identification of this problem was made by analysing the distribution of the intensity of the measured force, which indicated appreciably higher forces on the row of organs at the back (ch1, ch17, ch18, see Figure 6) in relation to those measured on the first row of organs behind the tractor (ch4, ch23, and ch24, see Figure 5). The main reason and validation of the suspicion were found by analysing the images in [42]. This deficiency partially generates the non-uniformity of the working depth, treated in Section 2.1.4.

Another deficiency valid only in the case where the MCLS operates in exploitation mode in any of its work variants is unsatisfactory soil fragmentation. According to the analysis in the field and on the photos, it is necessary to couple some shredding elements (discs, rollers, etc.) to the back of the structure (the granulometric analysis of the processed soil was not performed). In the research regime of the pure effects of the working body, the installation of additional shredding structures is prohibited because it alters the desired effects.

Another category of possible deficiencies, which can be avoided partially at the design stage and through systematic experimentation in working mode, are resonances and interferences, also called knocks. Both phenomena are defined in the literature, for example [59]. A search for possible resonances in the supports of the working organs is presented in Section 2.1.4 when calculating the frequency spectra for the twelve supports of the working bodies. The literature includes the results of some studies related to the problems of vibrations in agricultural machines [25,26,34,60–67].

### 3.4. Effects on the Quality of the Working Depth

The vibration of the supports of the cultivator's working organs involves relative movements horizontally, vertically, and laterally. The vibrations of the cultivator are complex; they are not reduced to the elastic vibrations of the working supports; there are also random rigid vibrations of the supporting structure and the working organs. All these types of vibrations have consequences for the main parameters of the work quality: working depth, initial working width, energy consumption, productivity, and comfort of the tractor driver.

In order to give only an ideal picture of the variations in the working depth due to random vibrations, we performed an elementary and ideal calculation on the deformations of the supports of the working bodies in the work process by applying some forces included in the experimental range. Using the finite element method and the model to calculate the first five natural frequencies, we estimated the relative displacements (deformations) in

the three directions for the extremity of the support to which the working body is attached. The results are given in Table 2.

**Table 2.** The values of the relative displacements of the tip of the support of the working body are calculated using a linear-elastic structural model built by the finite element method.

| Force Magnitude, N | Relative Displacement (Deformation), m | | |
|---|---|---|---|
| | $Ox$ (Lateral) | $Oy$ (Forward) | $Oz$ (Vertical) |
| 100 | 0.001060 | 0.01329 | 0.007152 |
| 200 | 0.003340 | 0.04182 | 0.023520 |
| 500 | 0.009401 | 0.01179 | 0.072160 |
| 1000 | 0.020660 | 0.25910 | 0.190700 |

The results in Table 2 are slightly exaggerated because the support material is assumed to be perfectly linearly elastic, which is not true. However, they suggest a good value as an order of magnitude for the relative displacements. The dominant overshoots are manifested in the forward and vertical directions on the ground. These values are approximately linearly related. At the average value of the force from the recordings related to the analysed experiment (500 N), the relative vertical displacement reaches 7 cm, which seriously affects the programmed working depth (10 cm). These situations are not frequent, but they are facilitated by exceeding a working speed limit, deformations of the working surface (unevenness), and errors in the working depth regulation system.

In such conditions, the cultivator cannot be a tool to process the soil with high precision (a maximum error of the order of 10% of the theoretical depth) at the working depth. A previously levelled and sufficiently crushed work surface, combined with an adapted work speed, are mandatory conditions for precise work in terms of working depth. A general picture of the quality of the work performed in terms of working depth is shown in Figure 21.

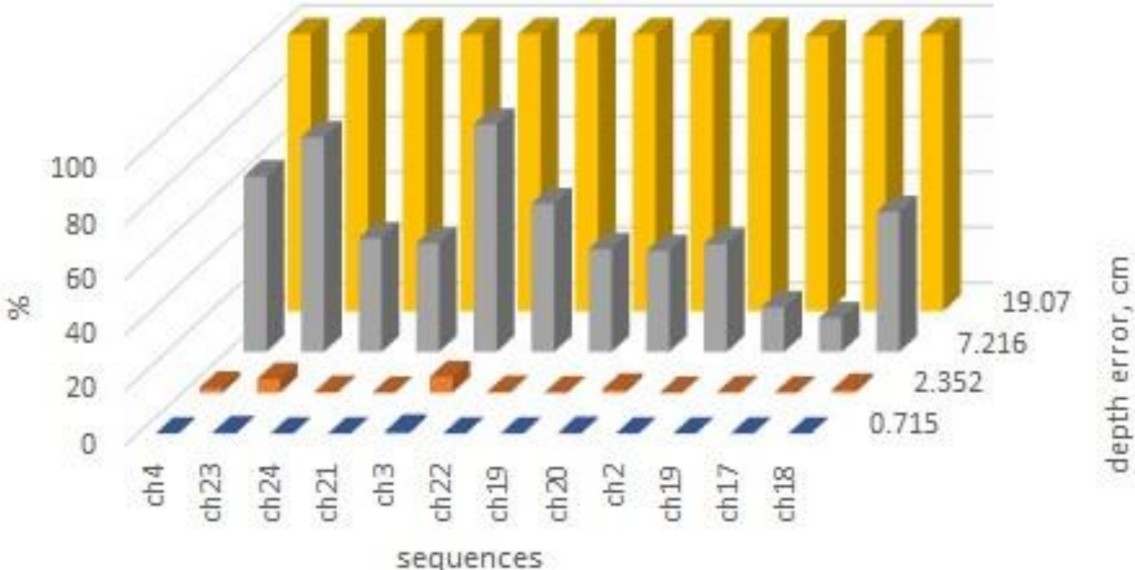

**Figure 21.** Errors in the working depth are calculated according to the measurements made on each channel.

The calculation from this subsection is based on the graphic representation in Figure 21, which uses the hypothesis of the operation of the supports of the working bodies of the MCLS in a linear-elastic regime without damping, which leads to an overestimation of the relative displacements. Direct measurements regarding the working depth and its

monitoring are insufficiently developed in the available experimental techniques; therefore, the estimation method described above was used.

An option to Improve the accuracy of the working depth is to ballast the load-bearing structure by adding some ballast materials. Increasing the precision of the working depth in this way will lead to higher energy consumption and, possibly, a decrease in productivity.

Additionally, a problem that must be solved for this machine is finding the maximum speed at which it can work without manifesting intense vertical oscillations throughout the entire structure, which completely compromises the quality of soil processing. The speed limit, which is determined through tests, is dependent on the characteristics of the soil, so an exact speed limit cannot be indicated a priori.

## 4. Comments

According to [51], most random vibration analyses are intended to realistically characterise the behaviour of structural systems excited by random inputs. The responses of real systems are known only when measured during their actual physical loading (and even then, only approximately). The effects caused by the use of simplifying assumptions in the process of numerical simulation of physical systems are rarely evaluated. Because of the depth of our knowledge in the analysis of linear systems subject to stationary media, we often idealise real systems as linear and the inputs as stationary. Practically, all real systems are nonlinear and random to a small or large extent. Therefore, the characterisation of the response can only be approximate. Such reasoning also directed our attention to the experimental or theoretical-empirical research of the working processes of agricultural machines.

Starting from these realities, we tried to formulate the problem within the strict framework of random vibrations. We tried to use as few simplifying assumptions or assumptions from the theoretical field of linear vibrations as possible. For this, we constructed the characteristic functions of the empirical sequences exactly as they result from the definition (Section 2). Probability densities and probabilities are obtained directly by using the numerical sequences and interpolating them with spline functions. We did not use probability density modelling with idealised functions (normal distributions, Student, Fischer, etc.), and consequently, the resulting probabilities are also modelled on real data.

Representations of the spectral frequency distributions of the twelve supports of the working bodies (Section 2.1.4, Figures 15 and 16) can be found in [24] for the chassis of a harvester. The orders of magnitude and the numerical values are also comparable, being characteristic of the working processes of agricultural machines. As values, the forces resulting from the experimental study described in this article are comparable to those found for the working bodies and also for a cultivator by the authors [68]. The obtained frequency spectra are similar to those from [69]. Unfortunately, we could not get accelerometer signals from the experimenter, which was very important. Acceleration recording remains a problem to be solved for MCLS. Some of the forces' values and the frequency spectra's frequencies were also obtained in [41]. In the same register of concerns and spectral values, [26] estimates the effects of frequencies and vertical accelerations on several types of tractors using frequency-acceleration diagrams. Such diagrams were not accessible for this paper, so frequency-load force diagrams on the working body were used instead. However, the principle of selecting deficient or dangerous work regimes is the same. The authors [29] use the same principle of spectral selection for the vibration level on the front axle of the Valtra 800 L tractor. As an estimator of the vibration level, the authors [29] use RMS (root mean square) amplitude [70] for the experimentally recorded acceleration sequences in the version indicated by the authors [29]. The RMS maximums are recorded in the frequency range where the maximum oscillation amplitudes of our force sequences were located. The optimisation of the frame of a precision agricultural machine for sowing vegetables by avoiding resonances, based on the same principles as in this article, is described in [61]. The authors proceed similarly [36] for cultivators with working bodies very similar to those with which the MCLS cultivator was equipped in our

experiments. Similarly, the authors [62] proceed with the study of the vibration of rapeseed seeds, additionally using high-speed photography and image recognition. All the frequency spectra (in this case normalised) are used in [67] to carry out experimental investigations with vertical damping for block-modular aggregates. As in this article, the authors [71] use spectral analysis and finite element analysis to predict fatigue accumulation in the arm of an excavator. The use of statistical analysis to estimate fatigue life dates back a long time [72], even for an agricultural cultivator; this is one of the main applications of the study of random vibrations. A similar approach for the testing of agricultural machines in an accelerated regime is proposed by the authors [73], in the same manner that we also proposed in Section 2.1.2. It is emphasised that not only the value of the load matters but also its frequency. The authors proceed similarly in [74,75]. A solution to reduce the force of resistance to advance for a cultivator by inducing forced vibrations is presented in [34]. Additionally, the return to some classical solutions, which had been partially abandoned, appeared after studies of random vibrations [76].

**5. Conclusions**

(C1) The analysis of random vibrations on any of the MCLS complex cultivator's variants is an opportunity and an obligation that must be mentioned in the work methodology that uses this equipment for research. The experiments carried out with the MCLS variants were also its first research applications. The literature shows that this is the most realistic way of approaching work processes in agriculture, having indisputable advantages over mathematical modelling and simulation. Among the most essential advantages is eliminating some assumptions that are often overly simplistic, lacking real motivation, and, above all, rarely verified, at least in the validation process. Moreover, simulation and mathematical modelling must validate their results experimentally; otherwise, they remain only scientific ballast.

(C2) The main results with immediate applications of random vibrations are the improvement of noise and vibration, the determination of dangerous demands for the elements of the structure, the determination of some limit values necessary for the choice of the material of some parts and sub-assemblies in the design, the estimation of the lifetime of the required elements in the regime of accumulation of fatigue, the identification of design or work regime deficiencies, and procedures for improving the quality of soil processing.

(C3) The presented results underline the deficiencies that must be resolved to increase the grower's performance. The recommendations refer to the control of the horizontalisation of the structure during work (with consequences for increasing the quality of soil processing relative to the working depth), the possibilities of ballasting the structure for better control of the working depth, limiting the operating speed, and the requirement of soil processing on some lands with satisfactory flatness.

(C4) A vital conclusion drawn from the results obtained is that to avoid malfunctions or damage due to premature fatigue, it is recommended that the "weakest" design elements, the ones that give way first, be among the cheapest or be simple, easy-to-replace, and cheap elements inserted into the subassemblies that will fail first in the event of an overload. We refer in particular to the fact that it is not rational to take into account the maximum loading for the structure (experimentally recorded). However, it is good to impose provisions for introducing safety elements. Thus, important and expensive subassemblies and parts will be protected. In the design of the 1960s and 1970s, for example, there were safety screws on ploughs that, by giving way first, protected the working bodies, their supports, and the load-bearing structure. Later, the abandonment of some of these safety elements produced the failure of load-bearing structures, for example.

(C5) Since we failed to validate the intuitive assumption that the working organs in the first line behind the tractor would be the most intensively requested, neither in these investigations nor in those described in [41], we deduce that, at least in the experimental conditions, the distribution of the maximum load is random over the twelve working bodies. An important consequence of this finding is that we cannot schedule a rotation

of the working bodies on the load-bearing structure positions to equalise the wear and use the full working capacity of all working bodies. Moreover, a statistically higher load was observed for the working body in the last line of organs in the load-bearing structure. First, only the result of the statistical analysis was validated based on the analysis of the images taken during the experiments. It is shown that certain aspects must be fixed in the operation of this machine, which will improve the quality of the work.

(C6) The results of the experimental research agree with those in the literature, presenting significant randomness and revealing average values of the efforts at the level of working bodies expected from a theoretical point of view. The experiments present an original character through a large number of measurement points, the study of the distribution of their results, and the investigation of the links between these measurements.

Obviously, in such a complex problem as soil processing, there will always be future work directions. In the case of the problem whose research is described in this article, there are many open problems, among which are mentioned: the dependence of the descriptive and inferential statistical characteristics on the work speed; the estimation of the soil processing problem with these work organs on all the work variants of the MCLS; solving the problems already listed for all the categories of working bodies that can be mounted on the load-bearing structure of the MCLS; and the study of the processing speed limit depending on the quality of the work performed. All these problems, which do not cover even half of the ways to continue the studies, should be carried out on several types of soil at different humidity levels, aiming, if not to optimise, at least to improve agricultural soil processing. Another interesting problem is the selection of a minimum number of measurements of the soil characteristics that can be used to fix the starting parameters of the soil processing. Among these measurements, we mention humidity, resistance to penetration, plant residue amount or density, and soil resistance's quantified characteristics (apart from resistance to penetration).

Before choosing a way to continue, evaluating the possible benefits of each possibility of continuing the research is required. The evaluation creates the conditions for an optimal choice of development paths, considering the material and human resources, especially the enormous costs of experimental research, which in the end necessarily also includes the theoretical one.

**Author Contributions:** Conceptualisation, P.C.,N.C. and V.M.; methodology, P.C.,N.C. and V.M.; software, P.C. and C.P.; validation, P.C., N.C., C.P., R.S., C.M.-I., O.-D.C. and E.-A.L.; formal analysis, P.C.,N.C., R.S., N.-V.V., C.M.-I. and E.-A.L.; investigation, P.C., N.C., C.P., V.M., N.-V.V., N.U., M.M., C.M.-I. and O.-D.C.; resources, P.C., R.S., N.-V.V., N.U. and M.M.; data curation, P.C, C.P., V.M., O.-D.C. and E.-A.L.; writing—original draft preparation, P.C. and N.C.; writing—review and editing, P.C., V.M. and N.U.; visualisation, N.U., O.-D.C. and E.-A.L.; supervision, P.C., V.M., N.-V.V., N.U. and M.M.; project administration, P.C., V.M., R.S. and C.M.-I.; funding acquisition, V.M., N.-V.V., N.U. and M.M. All authors have read and agreed to the published version of the manuscript.

**Funding:** The APC was funded by the National University of Science and Technology Politehnica Bucharest, Romania, within the PubArt Program.

**Institutional Review Board Statement:** Not applicable.

**Conflicts of Interest:** The authors declare no conflict of interest.

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
