# Peer review of "The Random Vibrations of the Active Body of the Cultivators"

_agriculture, doi:10.3390/agriculture13081565_

Round 1

Reviewer 1 Report

1. It is recommended to use a paragraph to describe random phenomena, randomness, and its measurement methods.

2. “In the range of demands considered, the vibrations can be considered linear-elastic. What is the range?

3. The measurement of the forces on each support of the working organs was done by the strain gauge methodology. The authors could provide some references to show how other scholars have dealt with similar problems and explain why you choose strain gauge methodology as your solution.

4. The conclusion should be related to the experimental results. The authors could draw a conclusion by analyzing the results of the experiment to make it more reasonable.

5. Format and writing errors:

(1) Reference 44 is not cited in this paper; there is a problem with the number of the formula in the text, and two formulas (1); it is recommended to use a three-line table; all text in the figure is recommended to use a unified font. (2) There are a large number of subjective sentences such as ' I, we ' in the text, which are suggested to be revised into third person.(3) The previous literature were not referenced appropriately. For example, On page 10, “The definitions of these notions can be found in all the literature dedicated to random vibrations, for example [18], [19], [21], [22], [23], [24], [25], [26], [27], and [28]”. Some of these references have the same authors. Therefore, there is no sense to list all of the author's paper. (4) Some caption were not put the same page with figure, For example, Fig. 9, The authors should correct these formatting issues to improve readability. Besides some figures in your paper are a bit blurry, For example, Fig. 13. Please consider replacing them with clearer ones.

Author Response

Thank you very much for your support

Response to Reviewer 1 in the attached file

Best regards,

Vergil Muraru

Reviewer 2 Report

The aim of this paper is to model the working processes of agricultural machinery. This is a very important aspect, given that these are often aspects that are not properly studied by manufacturers. The study is an in-depth study of a very important subject. It explores a subject that is particularly difficult to approach, both theoretically and practically. The research carried out is an interesting way of approaching agricultural work processes. It has advantages over mathematical modelling and simulation, even if it has some limitations. The methodology is well explained. The limitations of the methodology itself were also pointed out by the authors. In fact, they were asked to extrapolate them from the comments section and insert them in the comments section.

Figures 1 to 4 can be reduced in number.

figure 5 can be deleted.

The research work is interesting and well conducted and is also presented in excellent form.

I would like to suggest a change to the chapter comments, extrapolating the part about the research limitations and inserting them in materials and methods as 'research limitations"

Minor editing of English language required

Author Response

Thank you very much for your support.

Response to Reviewer 2 in the attached file

Best regards,

Vergil Muraru

Reviewer 3 Report

Abstract:

1. The first sentence assumes readers know a previous study.

2. What is 'MCLS'? needs to be clarified

3. no material and method is mentioned in the abstract

4. the abstract is lacking numerical results

Introduction

1.  Too lengthy, however, important concepts including cultivator design and analysis and random theory related to the study are missing

2. inappropriate and unnecessary direct quotation from literature in the first few paragraphs

3. Objectives are missing

Materials and Methods

1. No need of using 4 figures (Figs 1-4) to show background information

2. how the soil compaction level was measured? CPT?

3. Figure 5 is confusing, the detailed enlarged view should be outside of the original image

The subject of research 

1. most of the content in this section can be moved either to Methods or Results sections

2. working organs?

Extensive editing is needed

Author Response

Thank you very much for your support

Response to Reviewer 3 in the attached file

Best regards,

Vergil Muraru

Round 2

Reviewer 3 Report

Thanks for the effort but I cannot agree with some of the rebuttal without solid reasons.

No significant language improvement can be found. 

Author Response

Dear Sir,

Thank you very much for the second revision.

Best Regards,

Vergil Muraru
